# Leveling Control of Hillside Tractor Body Based on Fuzzy Sliding Mode Variable Structure

He Peng [1,2] , Wenxing Ma [1], Zhongshan Wang [1,3] and Zhe Yuan [1,*]

1 School of Mechanical and Aerospace Engineering, Jilin University, Changchun 130022, China; penghe888@126.com (H.P.); mawx@jlu.edu.cn (W.M.); wzssh_007@163.com (Z.W.)
2 Mechanical Engineering College, Beihua University, Jilin 132013, China
3 Jilin Agricultural Machinery Research Institute, Changchun 130012, China
* Correspondence: yuanzhe@jlu.edu.cn

**Featured Application: A sliding mode variable structure control algorithm based on fuzzy switching gain adjustment was adopted to achieve real-time dynamic auto-leveling control of a hillside tractor body. The tractor can remain within a $\pm2°$ tilting angle range during the leveling process, and can return to $0°$ after leveling, demonstrating good dynamic stability. High leveling accuracies can provide assistance with or in reference to obtaining solutions to the problems of tractor body leveling in hilly and mountainous areas.**

**Abstract:** To address the issues that arise when auto-leveling the vehicle body of a hillside tractor under complex working conditions, an auto-leveling control system was developed based on a newly developed hillside tractor and four-point body leveling mechanism. In this approach, leveling accuracy and stability were improved by adopting a sliding mode variable structure control algorithm based on fuzzy switching gain adjustment to achieve real-time dynamic auto-leveling control. To obtain curves of front and rear axle leveling displacement, speed, flow, pressure and body tilting angle during the leveling process, AMEsim/Simulink co-simulation was used to simulate and analyze the control system. The simulation results revealed that the tractor achieves a good leveling effect under complex working conditions in hilly and mountainous areas; the tractor can remain within a $\pm2°$ tilting angle range during the leveling process and can return to $0°$ after leveling, demonstrating good dynamic stability. To further assess the algorithm, a model of the system was submitted to live-testing on a custom-built auto-leveling test bench. Comparison of the test and simulation results revealed a close agreement between the two, indicating that the self-leveling control system and control algorithm developed in this study have high leveling accuracies. The results reported in this paper could provide assistance with or in reference to obtaining solutions to the problems of tractor body leveling in hilly and mountainous areas.

**Keywords:** hillside tractor; vehicle body auto-leveling; fuzzy adjustment; sliding mode variable structure control





## 1. Introduction

China is a large agricultural country with a total of $1.22 \times 10^8$ ha of cultivated land, of which hilly and mountainous areas account for approximately 63.2% [1]. Owing to the complex terrain and diversified agricultural operations carried out in such regions [2], the tractor is a typical piece of agricultural machinery and equipment, which can be used for the field operation or transportation of other agricultural machinery, thereby forming a vital component of agricultural production. Maintaining the level of the tractor body during operation in hilly and mountainous areas is difficult, and it is easy to overturn [3]. Therefore, conventional agricultural tractors are not widely used in agricultural production in hilly and mountainous areas. At present, several, new four-wheel or crawler tractors

have the ability to work safely in hilly and mountainous areas with small slopes. The main leveling methods include reducing the center of gravity, manual leveling and automatic leveling. The hillside tractors with automatic leveling have the best working performance. Automatic leveling can maintain the level of the body in real time, and improve the stability of hillside tractors during operation. However, the rapidity, stability and accuracy of leveling continues to require improvement [4]. Therefore, hillside tractors require a fast acting and high-precision self-leveling system to adjust vehicle body posture in a timely and effective manner to ensure that the vehicle level remains within an allowable range [5].

In recent years, vehicle body self-leveling systems have begun to be developed. Pijuan et al. designed a height-adjustable suspension mechanism that could level a chassis during vehicle movement to improve the obstacle-surmounting ability of the overall vehicle [6]; however, the system had a low leveling accuracy, which limited its application in tractors operating in hilly regions. In 2014, Alleyne and John Deere jointly developed a two-degree-of-freedom (2DOF) controller for combine harvester header height control based on the optimization of the control system. Simulation and test results revealed that a 2DOF controller mounted on a harvester header outperformed an independent feedback controller in terms of leveling, but that the leveling actuator had a delay, which lead to limitations in the application of the feedback control of header height [7]. Northwest Agriculture and Forestry University developed a hillside crawler tractor with an automatic leveling function; however, random noise generated by body vibration resulted in a less-than-ideal filtering effect, resulting in a body automatic leveling accuracy of only 40% [8]. Xu Feng et al. of Jilin Agricultural Machinery Research Institute studied a hillside tractor leveling system whose dynamic stability needed to be improved [9]. Shanghai Jiaotong University and Shandong Wuzheng Group jointly developed a four-wheel drive tractor with leveling functionality, in which a swing mechanism was installed between the rear drive axle and wheels to enable the horizontal leveling of the vehicle body through the height difference generated by the side-to-side swing of the mechanism; however, the actual operational effect of this assembly awaits verification [10].

In view of the facts that the current generation of hillside tractor auto-leveling systems have low leveling accuracy under complex working conditions and most of them can only function in a static state, this paper uses key national research and development plans to propose an auto-leveling system based on a new type of hillside tractor and four-point leveling mechanism. The proposed mechanism adopts a sliding mode variable structure control algorithm, based on fuzzy switching gain adjustment, to effectively improve leveling accuracy and stability. To assess the proposed system, joint performance simulation calculation and analysis of the control system were carried out. In addition, a custom-built leveling test bench was used to live-test a model of the real system with the results used to compare and verify the performance accuracy of the control system with the theoretical analysis.

## 2. Leveling System Design and Control Strategy

### 2.1. Leveling System Design

Here, we explain the designed hillside tractor body four-point leveling system to clarify our method, process, and control system results.

The tractor motion model subjected to a non-horizontal road surface is established as shown in Figure 1a. In this figure, the body wheelspan is $a$, the body wheelbase is $b$, $\lambda$, $\mu$ are the distance coefficients from the center of mass to the origin of the coordinate system, the horizontal plane coordinate system of the body in the initial position is $O\text{-}X_0Y_0Z_0$, the coordinate system of the body in the non-horizontal position becomes $O\text{-}XYZ$, the rotation angle of the coordinate axis $X$ is $\alpha$, and the rotation angle of the coordinate axis $Y$ is $\beta$.

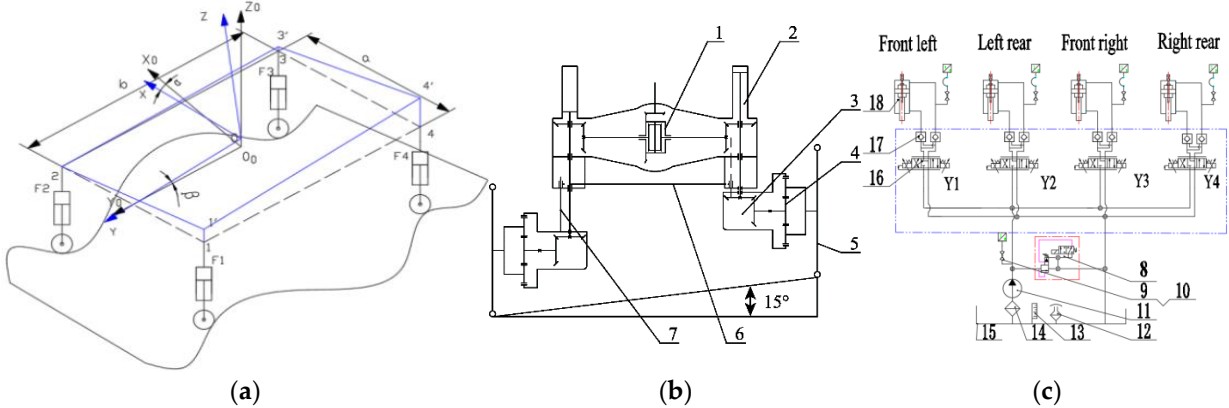

(**a**)             (**b**)             (**c**)

**Figure 1.** Principle, structure and hydraulic system of leveling drive axle: (**a**) The motion model on non-horizontal road surface; (**b**) Three-dimensional structure; and (**c**) Hydraulic system. 1 central transmission assembly; 2 leveling cylinder assembly; 3 steering knuckle assembly; 4 wheel transmission assembly; 5 tire; 6 steering rod; 7 guide pillar; 8 overflow valve; 9 pressure measuring joint; 10 pressure gauge; 11 gear oil pump; 12 hydraulic air filter; 13 liquid level thermometer; 14 oil suction filter; 15 oil tank; 16 electromagnetic proportional directional valve; 17 hydraulic operated check valve; 18 leveling cylinder.

For the non-horizontal state, the functional relationship between the force $F$ of the rodless cavity of each hydraulic cylinder and the load gravity $G$ is,

$$A \cdot F^T = B, \tag{1}$$

where,

$$A = \begin{bmatrix} 1 & 1 & 1 & 1 \\ a/2 & a/2 & a/2 & a/2 \\ b/2 & b/2 & b/2 & b/2 \\ l_1 & -l_2 & l_3 & -l_4 \end{bmatrix}, F = \begin{bmatrix} F_1 & F_2 & F_3 & F_4 \end{bmatrix}^T, B = \begin{bmatrix} G \\ \lambda a \cdot G \\ \mu b \cdot G \\ 0 \end{bmatrix}.$$

Because the initial state of the leveling hydraulic cylinder is the complete recovery state of the cylinder rod, the highest level of the body support point is the target point after excitation from the hilly slope road surface. After analysis and calculation, the displacement of each support point of the tractor is obtained.

$$\begin{cases} l_1 = b \cdot \sin\beta\cos\alpha \\ l_2 = a\sin\alpha + b\sin\beta\cos\alpha \\ l_3 = a\sin\alpha \\ l_4 = 0 \end{cases} \tag{2}$$

Figure 1b shows the working principle of the rear axle. The structure of the front axle is altered so that it has a kingpin tilting angle of 3° relative to the rear axle to obtain a front axle hydraulic cylinder stroke 5 mm larger than that of the rear axle hydraulic cylinder; otherwise, the leveling principle is the same for both.

Figure 1c shows the leveling hydraulic system, which supplies oil through a gear oil pump and stabilizes pressure through a relief valve. By changing the working state of the electromagnetic proportional valve, the corresponding leveling hydraulic cylinder can be extended or retracted [11–13]. When the tractor encounters a transverse slope during operation, the rodless cavity of the hydraulic cylinder on the lower side of the drive axle begins to charge, causing the hydraulic rod to stretch out to raise the lower side of the tractor and return the posture of the body to a horizontal state to ensure the safety and comfort of the ride.

### 2.2. Leveling Control Strategy

To improve the service life of the leveling system, no leveling is performed when the absolute value of the vehicle body tilting angle is less than 2°. When the tilting angle exceeds 2°, leveling is required to compensate for the displacement difference and ensure leveling accuracy. The specific leveling control strategy is shown in Figure 2.

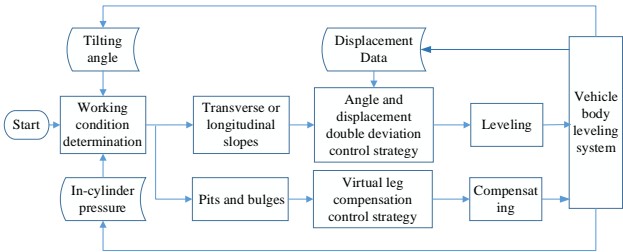

**Figure 2.** Leveling system control strategy.

Under actual conditions, a hillside tractor has two primary types of working state, that is, operation on transverse or longitudinal slopes, or on pits or bulges. The working condition should be comprehensively judged from the signals obtained from current body tilting and cylinder pressure data and a corresponding control strategy should be automatically selected. If one of the pressure values of the rodless chambers of the four leveling hydraulic cylinders suddenly drops or rises sharply, then the working condition can be determined to be a pit or a bulge; conversely, if the pressure values of the four leveling hydraulic cylinders show no obvious changes, but the absolute value of the tilting angle of the vehicle body is greater than 2°, then the working condition can be determined to be a transverse or longitudinal slope. Considering that the primary safety hazard facing a tractor working in hilly and mountainous environments is rollover, the control strategy involves leveling the vehicle body on horizontal slopes but not on longitudinal slopes.

A virtual leg compensation control strategy is adopted for pits and bulges. In this approach, an angle and displacement double deviation control strategy is applied for transverse slope leveling, as shown in Figure 3. The leveling control system takes an expected value of the tilting angle of the vehicle body of zero (that is, the attitude of the vehicle body is horizontal) as the input signal and the real-time tilting angle of the vehicle body; additionally, it takes the pressure of the rodless chambers of hydraulic cylinders and the displacement of the hydraulic cylinders as feedback signals. It controls the action of the hydraulic cylinders of the vehicle through the application of a leveling strategy and algorithm to compensate for the tilting angle of the vehicle body and maintain a level vehicle attitude.

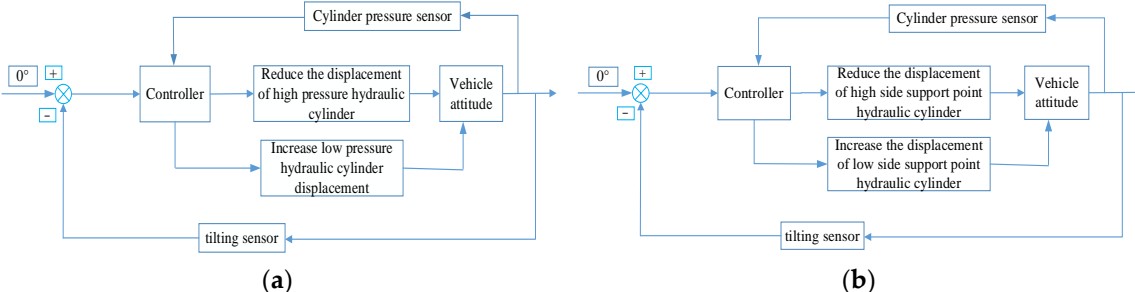

**Figure 3.** Double deviation control strategy of angle and displacement. (**a**) Virtual leg compensation control strategy. (**b**) Double deviation control strategy of angle and displacement.

### 3. Leveling System Model and Controller Design

*3.1. Construction of Mathematical Model of Body Leveling Hydraulic System*

The electro-hydraulic servo system of the proposed hillside tractor auto-leveling mechanism is a four-way-valve-controlled asymmetric hydraulic-cylinder position tracking system. Considering the internal and external leakage of the system, the control principles of the four leveling cylinders can be set to be identical, and a power model of the left rear leveling cylinder system can be expressed using the following three equations.

Valve flow equation,

$$q_L = k_q x_v - k_e p_L \tag{3}$$

where $k_q$ is the flow gain of the valve, m$^2$/s; $k_e$ is the flow pressure coefficient, m$^3$/(s·Pa); $x_v$ is the valve core displacement, m, and $p_L$ is the load pressure, N.

Flow equation of the hydraulic cylinder,

$$q_L = q_1 = A_1 \dot{y} + \frac{V_t}{4\beta_e} \dot{p}_L + C_{lp} p_L + C_f p_s \tag{4}$$

where $q_1$ is the flow rate of the rodless chamber of the hydraulic rod, m$^3$/s, $A_1$ is the area of the rodless chamber of the hydraulic cylinder, m$^2$, $y$ is the displacement of the hydraulic rod piston, m, $\beta_e$ is the equivalent elastic modulus of the oil, N/m$^2$, $V_t$ is the total effective volume of the hydraulic cylinder, m$^3$, $C_{lp}$ is the total effective leakage coefficient, and $C_f$ is the total additional leakage coefficient.

Force balance equation of the hydraulic rod,

$$A_1 p_1 - A_2 p_2 = m\ddot{y} + B_p \dot{y} + ky + F_L \tag{5}$$

where $A_2$ is the rod chamber area of the hydraulic cylinder, m$^2$, $m$ is the mass of the piston and the load converted to the piston, kg, $B_p$ is the viscous damping coefficient of the piston rod and the load, N·s/m, $k$ is the spring stiffness for the load, N/m, and $F_L$ is an arbitrary load force, N.

The dynamic characteristics of the valve-controlled hydraulic cylinder can be completely described using these three basic equations. By applying Laplace transformation and derivation simplification, the transfer function of the displacement of the leveling hydraulic cylinder $Y(s)$ and the displacement of the proportional valve spool $X_v(s)$ is,

$$G_4(s) = \frac{Y(s)}{X_v(s)} = \frac{\frac{k_q}{A_1}}{s\left(\frac{1}{w_h^2}s^2 + \frac{2\zeta_h}{w_h}s + 1\right)} \tag{6}$$

where $\zeta_h$ is the system damping ratio and $\omega_h$ is the natural frequency of the system.

The state equation of the system must be considered in designing a fuzzy sliding mode controller. For a system with input signals $r$ and output signals $y$, the deviation of the system will be $e = r - y$, and the error vector of the system can be defined as,

$$E = \begin{bmatrix} e_1 & e_2 & e_3 \end{bmatrix}^T = \begin{bmatrix} r - y & \dot{r} - \dot{y} & \ddot{r} - \ddot{y} \end{bmatrix}^T \tag{7}$$

According to the system model state space equation,

$$\begin{cases} y = x_1 \\ \dot{x}_1 = x_2 \\ \dot{x}_2 = x_3 \\ \dot{x}_3 = -w_h^2 x_2 - 2\zeta_h w_h x_3 + K_h w_h^2 u \end{cases} \\ \begin{cases} y = x_1 \\ \dot{x}_1 = x_2 \\ \dot{x}_2 = x_3 \\ \dot{x}_3 = -w_h^2 x_2 - 2\zeta_h w_h x_3 + K_h w_h^2 u \end{cases} \tag{8}$$

The error state equation can be written as,

$$\dot{E} = \begin{bmatrix} 0 & 1 & 0 \\ 0 & 0 & 1 \\ 0 & -w_h^2 & -2\zeta_h w_h \end{bmatrix} E + \begin{bmatrix} 0 \\ 0 \\ K_h w_h^2 \end{bmatrix} u + \begin{bmatrix} 0 \\ 0 \\ \ddot{r} + 2\zeta_h w_h \ddot{r} + w_h^2 \dot{r} \end{bmatrix} \tag{9}$$

### 3.2. Design of Fuzzy Sliding Mode Variable Structure Controller

### 3.2.1. Sliding Mode Controller Design

Sliding mode variable structure control has the advantages of high reliability and robustness. However, chattering occurs under the influence of uncertainties, such as interference, parameter changes and unmodeled dynamics. Scholars have proposed various weakening methods for the chattering phenomenon of the sliding mode variable structure [14,15]. The novelty of this paper lies in combining fuzzy control theory with the sliding mode variable structure control algorithm, and by applying it to the leveling control of tractors in hilly and mountainous areas, it can reduce the chattering phenomenon and show a good control effect on tractor leveling in hilly and mountainous areas.

The sliding mode function is initially designed according to the displacement tracking error, following which, the fuzzy regulator is designed to debounce the function, to replace $(e, \dot{e})$ with $(s, \dot{s})$ as the input signal, and to effectively evaluate the switching gain according to the sliding mode attainment condition and eliminate interference with the switching gain, thereby removing the chattering [16,17]. The principle of the controller operation is shown in Figure 4.

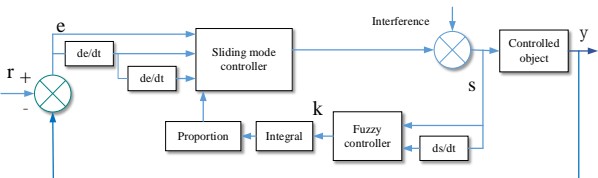

**Figure 4.** Controller principle.

The switching function of the sliding mode surface is defined as [18]

$$s = c_1 e_1 + c_2 e_2 + e_3 = c_1 e_1 + c_2 \dot{e}_1 + \ddot{e}_1 \tag{10}$$

where $e_1 = r - y$, $r$ is the desired output signal and $y$ is the actual output signal. By selecting the desired pole as $-34.4 \pm 168.52i$ to determine $c_1$ and $c_2$ through the pole configuration method, the parameters $c_1 = 29{,}582$, $c_2 = 68.8$ are obtained.

To ensure that $s\dot{s} < 0$, the law of the sliding mode controller is set as [19]

$$u = \frac{1}{b_0} [(c_1 - a_2)e_2 + (c_2 - a_3)e_3 + K(t)\text{sgn}(s)] \tag{11}$$

where $a_2 = w_h^2$, $a_3 = 2\zeta_h w_h$, $b_0 = K_h w_h^2$.

$K(t) = max|F(t)| + \eta$, $F(t) = \ddot{r} + 2\zeta_h w_h \ddot{r} + w_h \dot{r}$ and $\eta > 0$.

The Lyapunov stability analysis is carried out. If the function is V = $s^2/2$, then,

$$\dot{V} = s\dot{s} = s(c_1 e_2 + c_2 e_3 + (-a_3 e_3 - a_2 e_2 - b_0 u(t) + F(t))) \tag{12}$$

The control law (11) is introduced in (12) as follows,

$$\dot{V} = s(-K(t)\text{sgn}(s) + F(t)) \leq -\eta|s| \leq 0 \tag{13}$$

In the sliding mode control law, the switching gain value $K(t)$ is the cause of chattering. It is used to compensate the uncertainty $F(t)$ to ensure that the sliding mode existence

condition is satisfied. If $F(t)$ is time-varying, $K(t)$ should also be time-varying in order to reduce chattering. Fuzzy rules can be used to achieve changes of $K(t)$ based on experience.

3.2.2. Fuzzy Rule Design

The next step is to design the fuzzy rules of the switching gain, $K(t)$.

The condition of existence of a sliding mode is $s\dot{s} < 0$.

When the system reaches the sliding surface, it maintains a hold on it using the gain $K(t)$ to ensure that the system motion can reach a sliding mode surface whose value is sufficient to eliminate the influence of uncertain items and ensure that the sliding mode existence condition, $s\dot{s} < 0$, is established [20,21]. The two fuzzy rules are as follows.

If $s\dot{s} > 0$, $K(t)$ will increase;

If $s\dot{s} < 0$, $K(t)$ will decrease.

Using these fuzzy rules, a fuzzy system can be designed based on the relationship between $s\dot{s}$ and $\Delta K(t)$, where $s\dot{s}$ is designated as the input and $\Delta K(t)$ as the output.

The fuzzy set of system input and output is defined as follows.

$s\dot{s}$ = {NB NM ZO PM PB};

$\Delta K$ = {NB NM ZO PM PB}.

where NB is a large negative value, NM is a medium-sized negative value, ZO is zero, PM is a medium-sized positive value, and PB is a large positive value.

The input and output membership functions of the fuzzy system are shown in Figures 5 and 6.

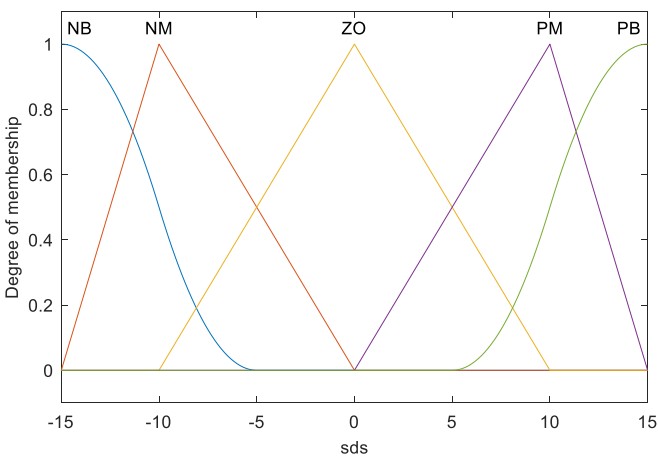

**Figure 5.** Membership function of fuzzy input variables.

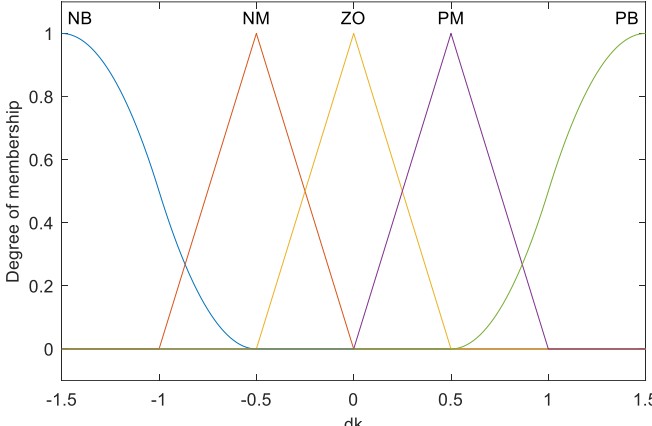

**Figure 6.** Membership function of fuzzy output variables.

The fuzzy rule design is as follows in Table 1.

**Table 1.** The fuzzy rule.

| Number | $s\dot{s}$ | $\Delta K$ |
|---|---|---|
| 1 | PB | PB |
| 2 | PM | PM |
| 3 | ZO | ZO |
| 4 | NM | NM |
| 5 | NB | NB |

The upper bound of $\overset{\wedge}{K}(t)$ is estimated using the integral method.

$$\overset{\wedge}{K}(t) = G\int_0^t \Delta K dt \tag{14}$$

where $G$ is the proportional coefficient.

Replace $K(t)$ in the formula (11) with a $\overset{\wedge}{K}(t)$, the control law becomes

$$u = \frac{1}{b_0}\left[(c_1 - a_2)e_2 + (c_2 - a_3)e_3 + \overset{\wedge}{K}(t)\mathrm{sgn}(s)\right] \tag{15}$$

## 4. Simulation Analysis

A performance simulation analysis of the control system is carried out under a joint AMEsim and Simulink environment. The hydraulic simulation model of the self-leveling system is established in AMEsim, and the fuzzy sliding mode variable structure control algorithm is established in Simulink; the two parts interact through an interface block module to achieve joint simulation [22–24].

### 4.1. AMEsim Hydraulic System Model Building

The hydraulic system simulation model established in AMEsim is shown in Figure 7. The model includes four hydraulic cylinder leveling modules corresponding to the left front, left rear, right rear, and right front cylinders, respectively; an interface block module implemented in Simulink is used for integration. Following the establishment of the model, a mathematical model is selected for each module according to the actual needs of the sub-model mode, and then the parameters for each sub-model are set according to the parameters of the hydraulic components of the hilly tractor. The primary parameters are listed in Table 2.

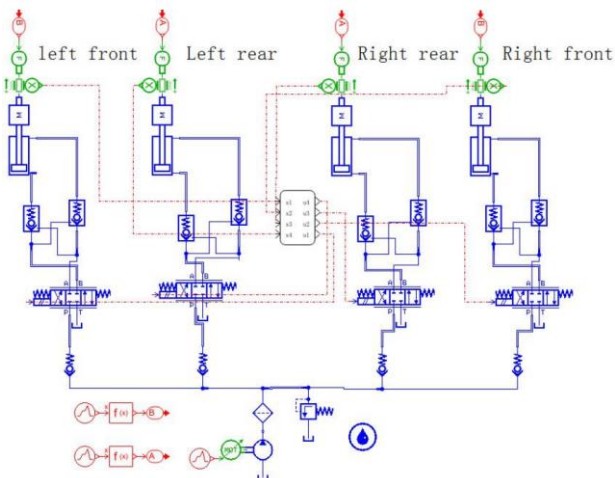

**Figure 7.** Hydraulic system simulation model of hillside tractor.

**Table 2.** Table of hydraulic component parameters.

|   | Parameter | Value |
|---|-----------|-------|
| 1 | Front axle hydraulic cylinder stroke | 286 mm |
| 2 | Rear axle hydraulic cylinder stroke | 281 mm |
| 3 | Pump | 14.6 mL/r 2000 r/min |
| 4 | The minimum used mass of the vehicle body | 1260 kg |
| 5 | Mass distribution ratio (front: rear) | 4:6 |
| 6 | Piston diameter | 63 mm |
| 7 | Oil density | 880 kg/m$^3$ (40°) |
| 8 | Electro-hydraulic proportional directional valve | 25 MPa 43 L/min |

### 4.2. Simulink Controller Model Construction

A MATLAB/Simulink control simulation model is constructed to reflect the leveling control strategy, as shown in Figure 8.

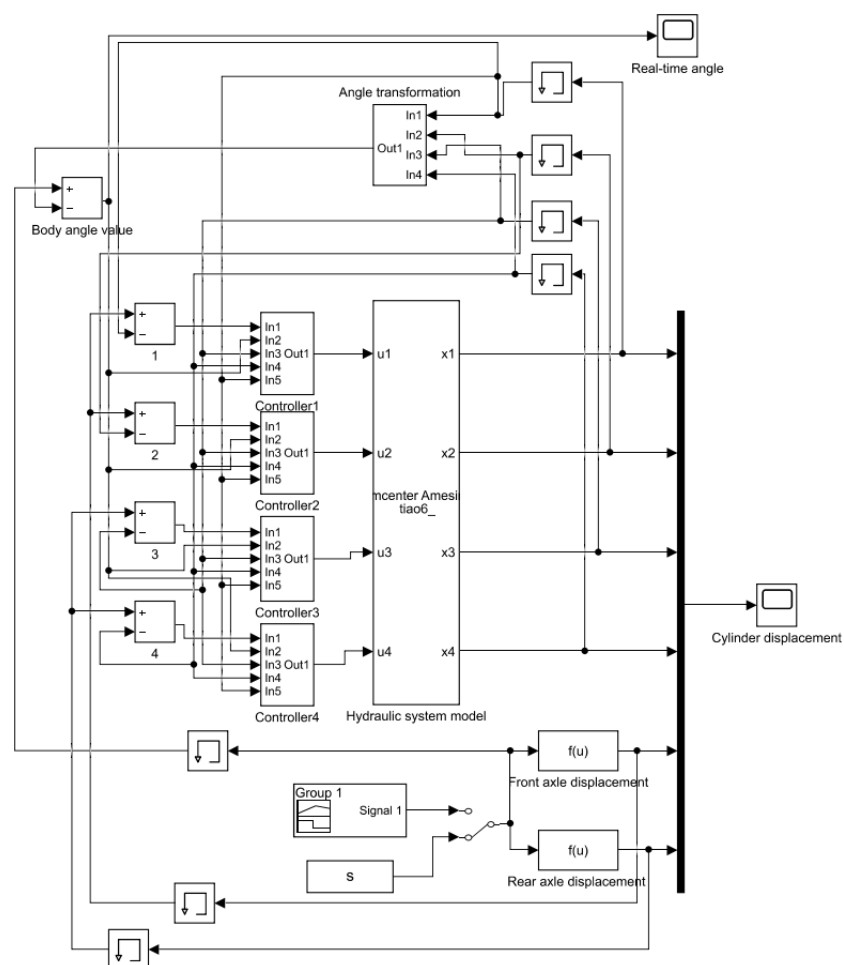

**Figure 8.** Control system simulation model of hillside tractor.

The model contains four controllers that provide different control signals to the respective hydraulic proportional valves following model computation. The controllers incorporate control strategies and a sealed fuzzy sliding mode variable structure control algorithm module (Figure 9a).

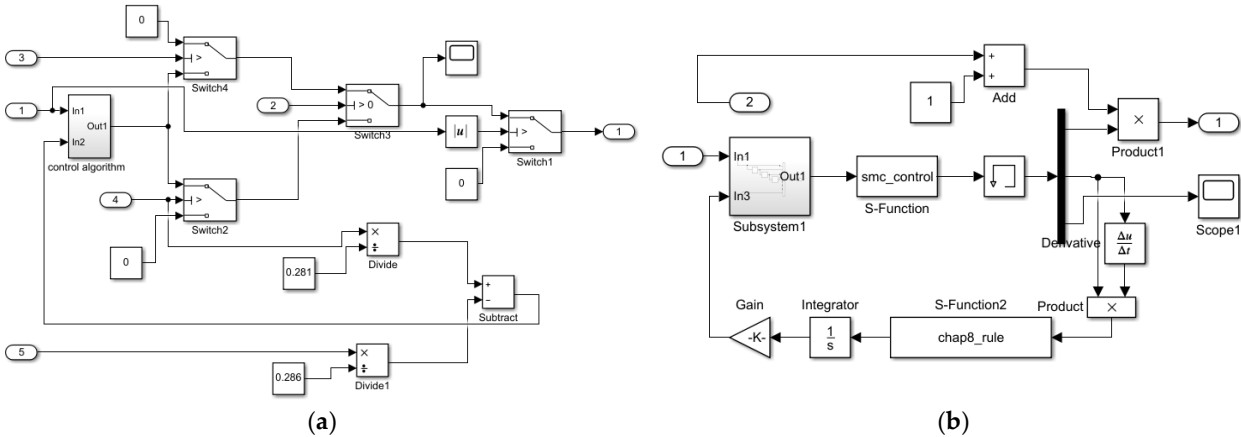

(a)  (b)

**Figure 9.** Control system simulation program packaging module. (**a**) Controller packaging module. (**b**) Fuzzy sliding mode variable structure arithmetic packaging module.

The specific structure of the fuzzy sliding mode variable structure control encapsulation module, which is shown in Figure 9b, comprises two parts: a sliding mode variable structure controller and a fuzzy regulator.

### 4.3. Simulation Result Analysis

4.3.1. Simulation Analysis of Spectrum Excited Leveling of Gravel Pavement with Transverse Slopes

During the leveling simulation, the spectrum of a gravel road surface collected from its transverse slope (Figure 10) is used for system excitation [25] with the left front and left rear of the hillside tractor used as the low side, and the right front and right rear as the high side. The simulation time is set to 80 s and the sampling frequency to 1000 Hz.

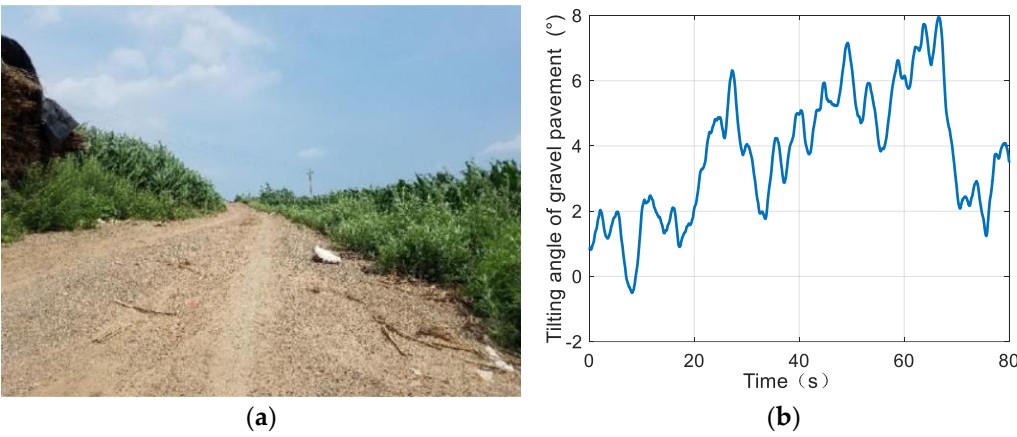

(a)  (b)

**Figure 10.** Variation of tilting angles of gravel transport pavement with transverse slopes. (**a**) Gravel transport pavement with transverse slopes. (**b**) Tilting angles of gravel transport pavement.

Joint simulation of the model with the sliding mode variable structure control algorithm based on fuzzy switching gain adjustment is applied to reveal the simulated displacement of the hydraulic cylinder leveling of the front and rear axles, as shown in Figure 11a, which shows that the simulated displacements of the front and rear axles of the hydraulic cylinders are essentially synchronous, and the lateral tilting of the vehicle body changes by approximately $\pm 2°$ (Figure 11b). As the displacement chattering generated by the sliding mode control of the hydraulic cylinder has been eliminated through the application of fuzzy switching gain adjustment, the tilting of the angle of the vehicle body under chattering is not evident, thereby ensuring the safety and dynamic stability of the tractor during operation.

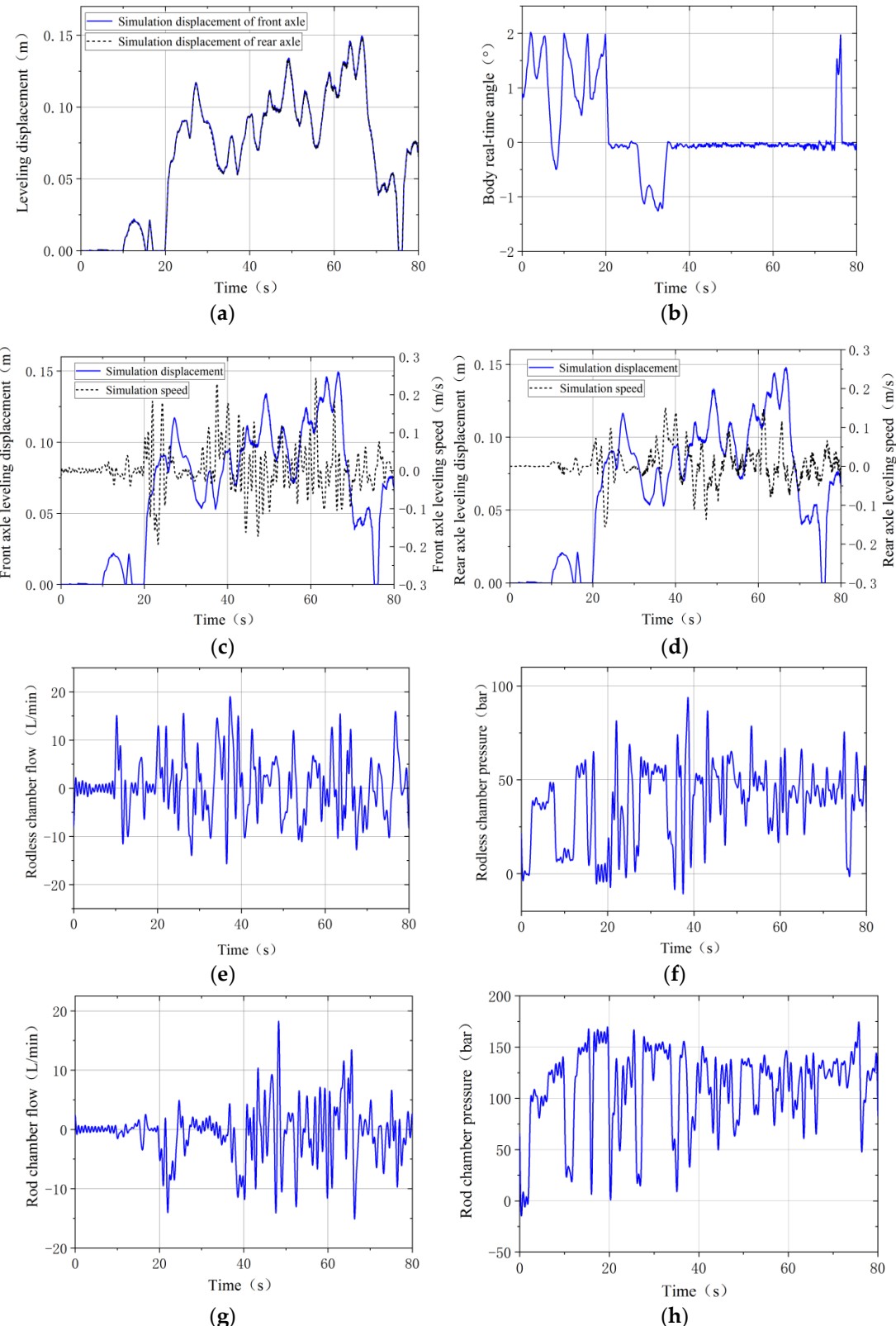

**Figure 11.** Characteristics of spectrum excitation leveling parameters on gravel pavement. (**a**) Leveling displacements of front and rear axles. (**b**) Body real-time angles. (**c**) Displacement and speed of front axle hydraulic cylinder. (**d**) Displacement and speed of rear axle hydraulic cylinder. (**e**) Flow rate of rodless chamber of rear axle. (**f**) Pressure in rodless chamber of rear axle. (**g**) Pressure in rod chamber of rear axle. (**h**) Pressure in rod chamber of rear axle.

Figure 11c,d, show, respectively, the characteristic curves of leveling displacement and variation in the velocities of the front and rear axles. As the leveling displacement of the front axle is larger than that of the rear axle, the speed of the front axle is higher than that of the rear axle during hydraulic cylinder extension and retraction; the maximum leveling speeds of the front and rear axles are approximately 0.2 m/s, indicating the fast leveling speed of the vehicle and, as a result, the leveling accuracy of the overall vehicle body.

Figure 11e,f, show, respectively, the flow and pressure characteristic curves for the rodless chamber of the rear axle hydraulic cylinder, whereas Figure 11g,h show the corresponding curves for the rod chamber of the rear axle hydraulic cylinder. It is seen that, even though the flow rate of the rodless chamber is greater than that of the rod chamber, the pressure in the former is lower than that in the latter. Nevertheless, both the flow rate and pressure values change within the safe range of the hydraulic system, that is, they are able to meet the requirements of the system, ensuring that the leveling is safe and reliable.

4.3.2. Simulation Analysis of Spectrum Excited Leveling of Transverse Slope Field Paths

A leveling simulation is carried out using collected field operation paths over transverse slopes (Figure 12) as the system excitation parameters [25]. In the simulation, the left front and left rear sides of the hillside tractor are set as the low side and the right front and right rear are set as the high side. The simulation time is set to 100 s and the sampling frequency to 1000 Hz.

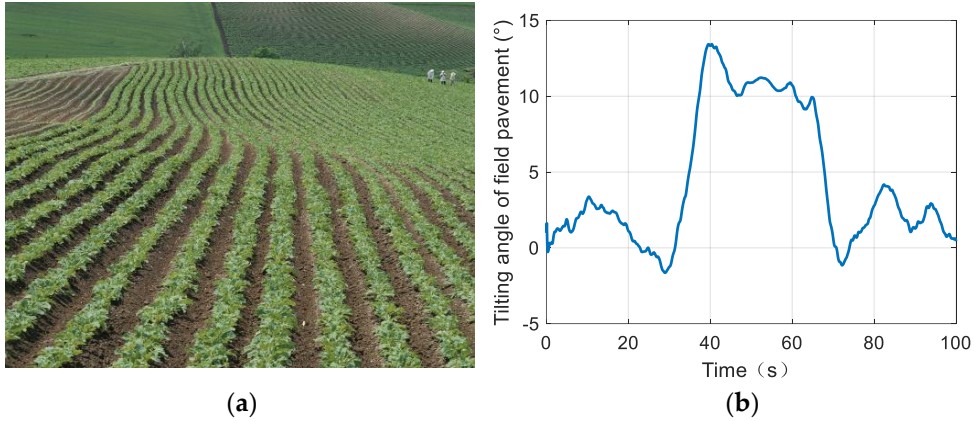

|  (**a**)  |  (**b**)  |

**Figure 12.** Variation of tilting angles of field pavement with transverse slopes. (**a**) field pavement with transverse slopes. (**b**) Tilting angles of field pavement.

The proposed sliding mode variable structure control algorithm based on fuzzy switching gain adjustment is used to carry out joint simulation to obtain leveling simulation displacements of the front and rear axle low-side hydraulic cylinders. As shown in Figure 13a, these are essentially synchronous, and as shown in Figure 13b, the tilting angle varies continuously within a $\pm 2°$ range. Under these conditions, the tractor is free from the danger of rollover; in addition, no obvious chattering is presented in the lateral tilting of the body, and dynamic stability of the leveling is illustrated.

Figure 13c,d show the characteristic curves of the leveling displacement and speed variation of the front and rear axles of the tractor, respectively. During the process of extension and retraction of the hydraulic cylinder, the maximum operating speed is 0.08 m/s, indicating the high precision and stability of the auto-leveling during field operations.

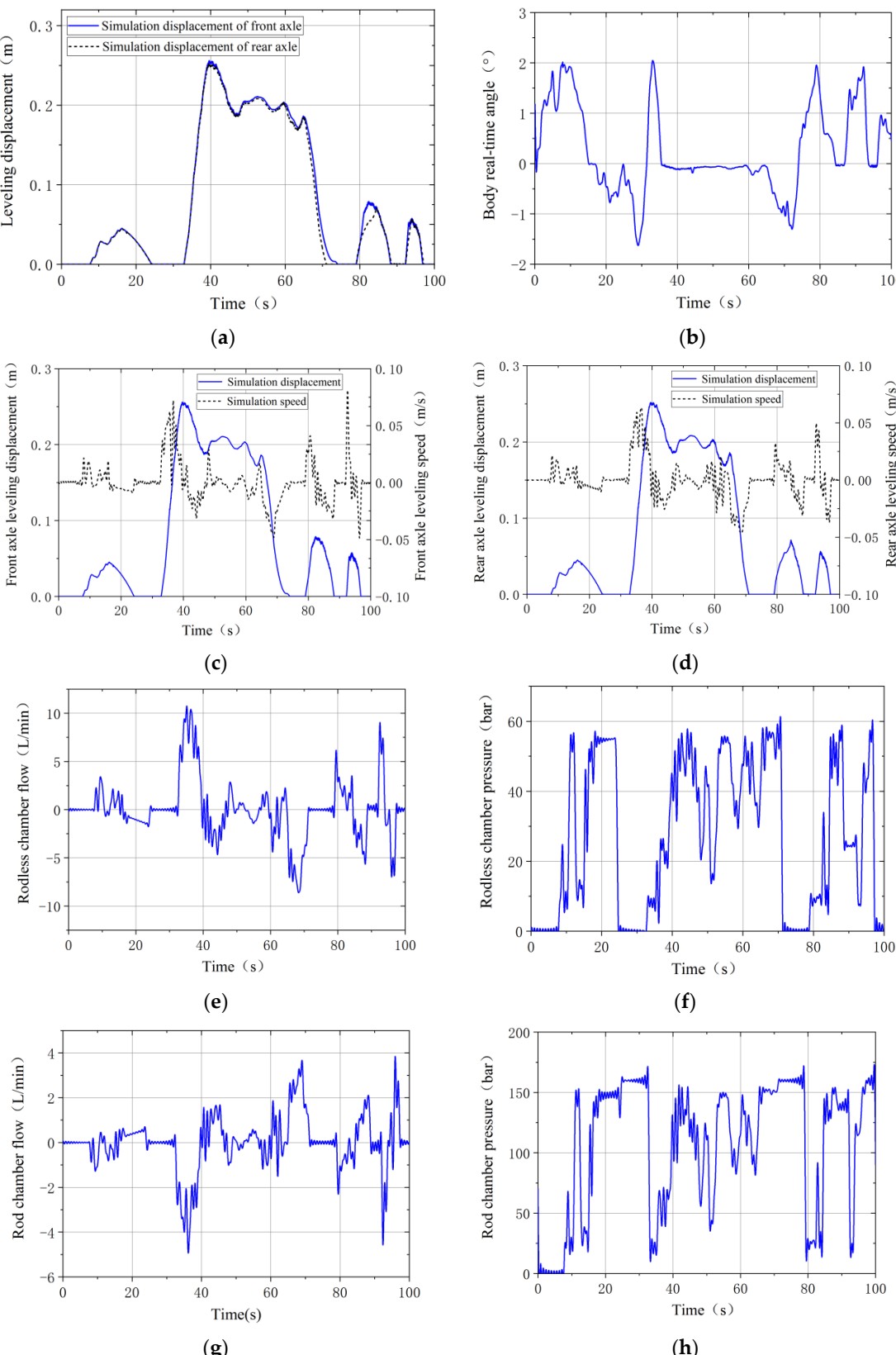

**Figure 13.** Leveling parameter characteristics of field pavement spectrum excitation. (**a**) Leveling displacement of front and rear axles. (**b**) Body real-time angles. (**c**) Displacements and speeds of front axle hydraulic cylinders. (**d**) Displacements and speeds of rear axle hydraulic cylinders. (**e**) Flow rates of rodless chambers of rear axles. (**f**) Pressures in rodless chambers of rear axles. (**g**) Flows in the rod chambers of the rear axles. (**h**) Pressures in the rod chambers of the rear axles.

Figure 13e,f show the flow and pressure characteristic curves of the rodless chamber of the rear axle, respectively, whereas Figure 13g,h show the corresponding curves for the rod chamber of the rear axle. The flow rate is 0 L/min when the hydraulic cylinder is not in action, and upon commencement of the leveling movement and hydraulic cylinder extension, the flows in the rodless and rod chambers become positive and negative, respectively, and when the hydraulic cylinder retracted, the flows in the rodless and rod chambers become negative and positive, respectively. These results correspond with the characteristics of the model and are within the safe range of the hydraulic system, demonstrating the safe and reliable leveling of the proposed tractor system.

## 5. Test Analysis

An auto-leveling test bench (Figure 14) is built to test the tractor's body structure and control system in hilly and mountainous conditions. The reliability and accuracy of the fuzzy sliding mode control algorithm are also verified in this test environment.

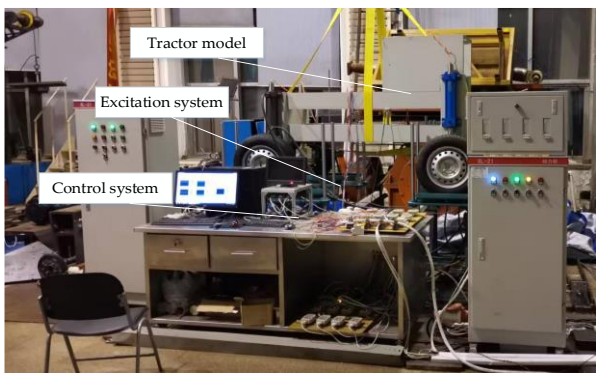 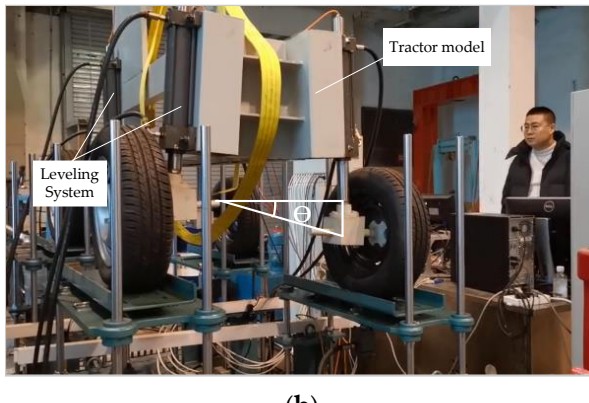

(**a**)  (**b**)

**Figure 14.** Vehicle body auto-leveling tests. (**a**) Vehicle body auto-leveling test bench. (**b**) Leveling state.

### 5.1. Composition of Auto-Leveling Test Bench

The auto-leveling test bench comprises three parts: a road spectrum excitation system, control system, and simulated vehicle body (Figure 14a). The spectrum excitation system can simulate two types of collected gravel transportation and field operation slope data; the control system utilizes an industrial computer that hosted the LABVIEW software to compile the sliding mode variable structure control algorithm program based on fuzzy switching gain adjustment, which in turn controls the angle of the drive axle leveling hydraulic cylinders. There are, in total, four leveling hydraulic cylinders on the front and rear drive axles of the simulated vehicle body, which are used to perform auto-leveling under the control system to verify whether the leveling capability meets the design requirements.

### 5.2. Auto-Leveling Tests

If the simulated vehicle body does not perform leveling motions when the road surface spectrum excitation system is functioning, the tractor body assumes a tilted position in which it is prone to rollover and the safety factor is low. Otherwise, if the simulated vehicle body can perform real-time leveling, in which the control system detects the tilting angle of the model vehicle body in real time through the tilting sensors installed on the body, and quickly calculates the leveling displacement to control the leveling hydraulic cylinder to perform corresponding leveling actions, then the simulated vehicle body remains in a horizontal state (Figure 14b) and ensures the safety of the driver.

### 5.3. Data Analysis of Auto-Leveling Tests

#### 5.3.1. Simulation Analysis of Spectrum Excited Leveling of Gravel Pavement with Transverse Slopes

The leveling tests is carried out using a sand and gravel transportation pavement spectrum of excitation obtained from a transverse slope (Figure 10). The spectrum is formed from the hydraulic cylinder leveling displacement, and velocity data and real-time vehicle body tilting data is collected by the displacement sensors installed on the hydraulic cylinder and the tilting sensors installed on the chassis of the vehicle body, respectively. The results are compared with the simulation data produced under the same conditions. The test and the simulation displacements exhibit the same trend (Figure 15), and the chattering is significantly improved by the fuzzy control, where it shows the fuzzy sliding mode control algorithm has a high-leveling accuracy. The fluctuation range of the test speed is larger than that of the simulation speed, and the real-time angle of the test fluctuates more significantly than that of the simulation, indicating an error in the mathematical modeling, which leads to the deviation from the simulation data. The test angle of the vehicle body is maintained within the $\pm 2°$ safe range, and the tractor will not roll over. The fuzzy sliding mode control algorithm has good leveling stability and can ensure the safety and stability of the tractor operating on the slope.

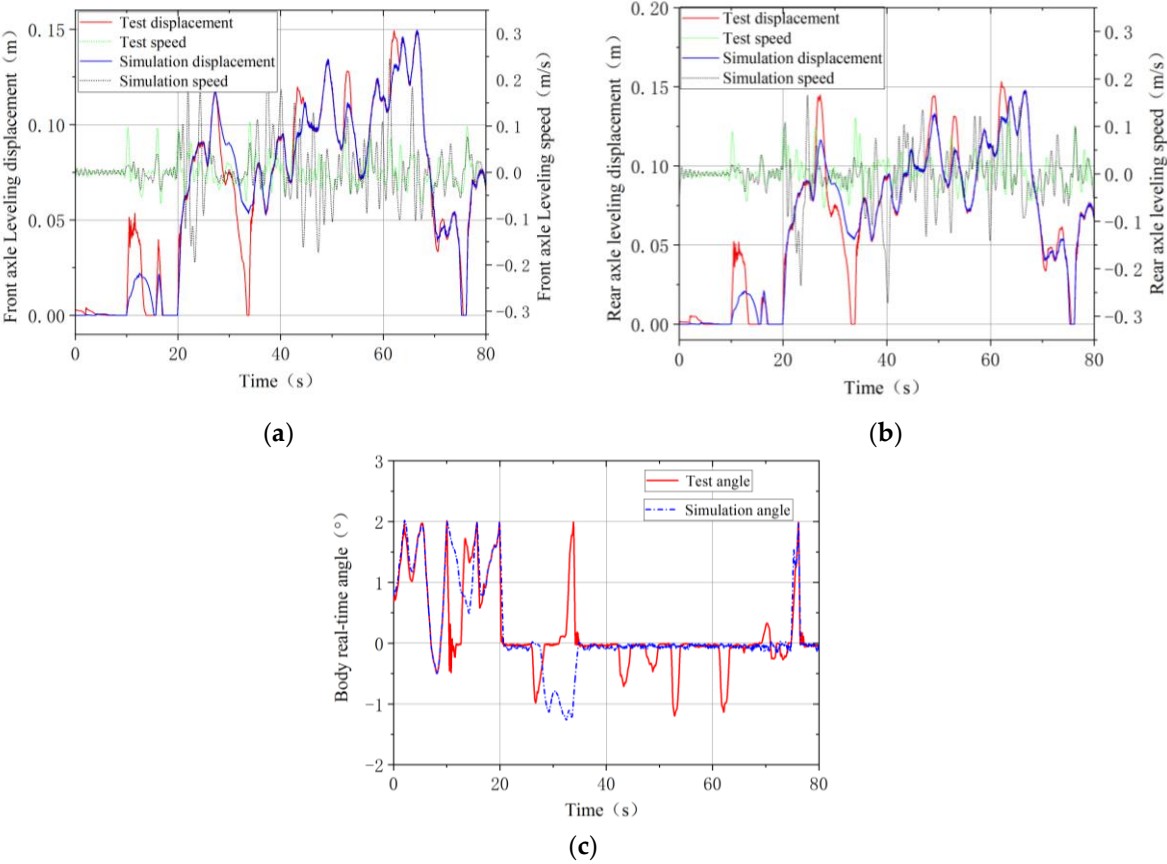

(a)

(b)

(c)

**Figure 15.** Comparison of spectrum excitation test and simulation data on gravel pavement. (**a**) Front axle displacement speed. (**b**) Rear axle displacement speed. (**c**) Vehicle body real-time tilting.

#### 5.3.2. Analysis of Horizontal Slope Field Spectrum Excitation Leveling Tests

Hydraulic cylinder leveling displacement, velocity data, and real-time angle data of the vehicle body collected from a field spectrum excitation leveling test on a transverse slope (Figure 12) are compared with simulation data obtained under the same conditions, as shown in Figure 16. The test and simulation displacements exhibit the same trend, and have a higher leveling accuracy; however, some deviation is present within the range of

10–25 s due to the errors in the mathematical modeling of the fuzzy sliding mode variable structure control algorithm. During operation, the tractor body inclination angle is within the $\pm 2°$ safe range, then the chattering is not significant, the fuzzy sliding mode control algorithm has good leveling stability, and the tractor body leveling is safe and reliable.

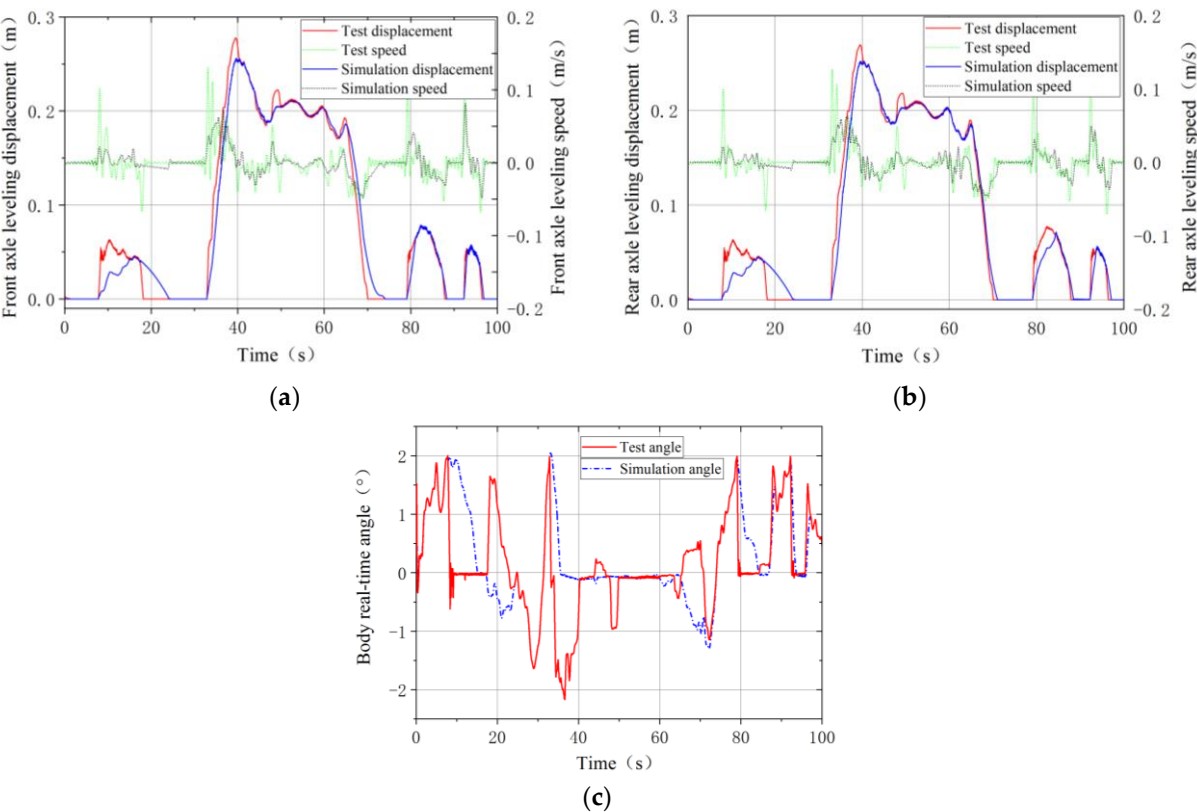

**Figure 16.** Comparison of field pavement spectrum excitation test and simulation data. (**a**) Front axle displacement speed. (**b**) Rear axle displacement speed. (**c**) Vehicle body real-time tilting.

A comparison of the body leveling test and simulation results obtained under the two excitation conditions, indicates that the test displacements are essentially consistent with the simulation displacements and that the simulated and test vehicle body angle changes are essentially the same, which verifies the correctness, accuracy, and reliability of the auto-leveling control system and its fuzzy sliding mode variable structure control algorithm.

## 6. Conclusions

This paper presented a vehicle body auto-leveling control system for hillside tractors and presented the results of simulation and testing using a custom-built auto-leveling test bench. The proposed fuzzy sliding mode variable structure control was simulated using software and tested using the auto-leveling system. Our primary conclusions are as follows:

(1) The proposed vehicle body leveling control system, which employs a fuzzy sliding mode variable structure control algorithm designed and built according to the functional requirements of the leveling system, is capable of real-time adjustment of controller parameters, guaranteeing a vehicle leveling effect.

(2) Data curves produced by measurements of front and rear axle leveling displacement, speed, flow, pressure, and body tilting angle during the tractor leveling process obtained through AMEsim/Simulink simulation reveal that the tractor has a good leveling effect under complex working conditions in hilly and mountainous areas.

The tilting angle of the tractor is kept within a $\pm 2°$ range during the leveling process and can return to $0°$ after leveling, indicating good dynamic stability.

(3) The proposed control method was used to carry out functional tests on an auto-leveling test bench, with the results confirming the correctness, accuracy, and reliability of the vehicle body auto-leveling control system and fuzzy sliding mode variable structure control algorithm. The test-confirmed results obtained in this study provide a theoretical basis for the design of auto-leveling control systems for hillside tractors.

(4) The tractor body leveling control of the sliding mode variable structure control algorithm, based on fuzzy switching gain adjustment, still exhibits some chattering and a singularity under an excitation signal with a large variation amplitude. Therefore, the control parameters can be optimized to further improve the body leveling response speed and control accuracy. Owing to the incomplete consideration of the influencing factors and parameters during the modeling process, a certain error is presented in the actual condition of the tractor leveling. If the actual model of the tractor can be used for the simulation in the follow-up study, the error of the electromechanical-hydraulic joint simulation can be reduced further.

**Author Contributions:** Conceptualization, H.P. and W.M.; methodology, H.P. and Z.Y.; software, H.P. and Z.W.; validation, H.P.; formal analysis, H.P. and Z.Y.; investigation, H.P. and Z.W.; resources, W.M.; data curation, Z.Y.; writing—original draft preparation, H.P.; writing—review and editing, W.M. and Z.Y.; visualization, H.P.; supervision, Z.Y.; project administration, Z.Y.; funding acquisition, W.M. All authors have read and agreed to the published version of the manuscript.

**Funding:** This work has been founded by the National Key Research and Development Program of China (2016YFD0700403).

**Institutional Review Board Statement:** Not applicable.

**Informed Consent Statement:** Not applicable.

**Data Availability Statement:** The data presented in the study are available on request from the corresponding author.

**Acknowledgments:** The authors would also like to thank the anonymous reviewers for their constructive comments.

**Conflicts of Interest:** The authors declare that they have no conflict of interest.

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
