# Peer review of "Leveling Control of Hillside Tractor Body Based on Fuzzy Sliding Mode Variable Structure"

_applsci, doi:10.3390/app13106066_

Round 1

Reviewer 1 Report

- Include rigorous stability analysis of the proposed control technique. - Please label Figure 14b so as to convey more information. - Many figures in the manuscript are not readable. Please increase their size as well as resolution. - Explicitly mention the novelty of the proposed Sliding Mode Control technique. - In Section 3.2.1, chattering reduction is mentioned. Refer to 10.3390/electronics12040876. - Transform the fuzzy rule design in the form of a Table. - Simulink diagrams (e.g. Figure 7b) is not giving any information and may be removed. Instead, draw a higher-level block diagram. - Having many references cited at one place is not usual e.g. [1-4], [5-8], [9-12]. - There are a lot of interesting and useful results/graphs, however with very limited discussion. Please extend the discussion on these graphs. - Include a brief theory of sliding mode control with reference to 10.3390/en12091669, 10.1201/9781420065619 and 10.1371/journal.pone.0281116 - Use appropriate error criteria like ITAE/IAE/ISE/ITSE for analysis. - Add 'Future work' in the conclusion section. - Please let the paper proofread by a native English speaker for linguistic improvements. - In Figures 2-4, for the sake of clarity, make the background colour of boxes as white.

Reviewer 2 Report

The introduction part of the Leveling Control of Hillside Tractor Body based on Fuzzy Sliding Mode Variable Structure article brings an sufficient overview of the former research in the area of interest.

Just check the formal way of the reference writing - there are missing spaces between the end of text and the reference brackets[].

The captions by the figure 1 could be formatted better, especially the explanation of the system parts 1 to 18 in the Fig. 1 c)

The formatting of the equations 3 to 9 should be improved too.

Please reference the equations 10 and 11, if possible.

Please check the explanations by fuzzy rules design description in 3.2.2, it the sentences used do not make any sense, please rewrite the rules description more understandable.

Check also the Fig. 8 and 9 captions.

It would be very helpful to put also the 3D model of the track to the chapter 4.3, if possible.

Please discuss also more the differences between the simulation and test results in Figs. 15 and 16.

Round 2

Reviewer 1 Report

Authors have addressed all the suggested changes.